# Nanosecond chain dynamics of single-stranded nucleic acids

Mark F. Nüesch [1], Lisa Pietrek [2], Erik D. Holmstrom [1,3,4] ✉, Daniel Nettels [1], Valentin von Roten [1], Rafael Kronenberg-Tenga[1], Ohad Medalia [1], Gerhard Hummer [2,5] ✉ & Benjamin Schuler [1,6] ✉

The conformational dynamics of single-stranded nucleic acids are fundamental for nucleic acid folding and function. However, their elementary chain dynamics have been difficult to resolve experimentally. Here we employ a combination of single-molecule Förster resonance energy transfer, nanosecond fluorescence correlation spectroscopy, and nanophotonic enhancement to determine the conformational ensembles and rapid chain dynamics of short single-stranded nucleic acids in solution. To interpret the experimental results in terms of end-to-end distance dynamics, we utilize the hierarchical chain growth approach, simple polymer models, and refinement with Bayesian inference to generate structural ensembles that closely align with the experimental data. The resulting chain reconfiguration times are exceedingly rapid, in the 10-ns range. Solvent viscosity-dependent measurements indicate that these dynamics of single-stranded nucleic acids exhibit negligible internal friction and are thus dominated by solvent friction. Our results provide a detailed view of the conformational distributions and rapid dynamics of single-stranded nucleic acids.

Nucleic acids and proteins have very different chemical compositions. However, as linear biopolymers, both can sample a myriad of chain configurations, and the resulting dynamics play an essential role in their folding and function. The chain dynamics of unfolded and disordered proteins have been characterized extensively with a broad range of methods[1–8] because of their importance for protein folding[9,10] and the behavior of intrinsically disordered proteins[11,12]. The chain dynamics of single-stranded nucleic acids (ssNAs) are less well characterized, despite their importance for many biological processes, particularly those associated with gene expression and RNA folding[13]. While double-stranded nucleic acids are very stiff, with persistence lengths in the range of tens of nanometers[14], ssNAs rapidly sample conformationally heterogeneous ensembles and exhibit persistence lengths in the 1- to 3-nm range[15–22]. Fluorescence quenching

experiments have yielded end-to-end contact rates of ~$10^6$ s$^{-1}$ for ssDNA with lengths from 2 to 20 nucleotides, demonstrating their pronounced flexibility and rapid dynamics[4,23–25]. However, the quantitative interpretation of contact rates in terms of chain dynamics requires detailed knowledge of both the distance dependence of the quenching process and the short-distance tail of the end-to-end distance distribution, which is very sensitive to local structure formation and steric accessibility[26–28]. Modeling the behavior of ssNAs with molecular simulations has also been more challenging than for proteins, primarily because it has been difficult to capture the subtle balance of interactions such as base stacking with sufficient accuracy[29,30].

To improve our quantitative understanding of chain dynamics in ssNAs, we used single-molecule Förster resonance energy transfer

[1]Department of Biochemistry, University of Zurich, Winterthurerstrasse 190, 8057 Zurich, Switzerland. [2]Department of Theoretical Biophysics, Max Planck Institute of Biophysics, Max-von-Laue-Straße 3, 60438 Frankfurt am Main, Germany. [3]Department of Chemistry, University of Kansas, Lawrence, KS, USA. [4]Department of Molecular Biosciences, University of Kansas, Lawrence, KS, USA. [5]Institute for Biophysics, Goethe University Frankfurt, 60438 Frankfurt am Main, Germany. [6]Department of Physics, University of Zurich, Winterthurerstrasse 190, 8057 Zurich, Switzerland. ✉e-mail: erik.d.holmstrom@ku.edu; gerhard.hummer@biophys.mpg.de; schuler@bioc.uzh.ch

(FRET) combined with nanosecond fluorescence correlation spectroscopy (nsFCS) to probe the long-range intramolecular dynamics of short homopolymeric single-stranded RNA (ssRNA) and DNA (ssDNA) oligonucleotides. To interpret the results in terms of distance dynamics, we combine them with distance distributions obtained from the recently developed hierarchical chain growth (HCG) approach[31], which produces structural ensembles that for RNA have been shown to be in good agreement with experiments, including nuclear magnetic resonance (NMR), small-angle X-ray scattering (SAXS), and single-molecule FRET[32]. We also compare the resulting distributions from HCG to distance distributions of polymer models commonly used to interpret single-molecule FRET data. Our results reveal exceedingly rapid chain dynamics of single-stranded nucleic acids. We observe no detectable internal friction, which indicates the absence of intrachain interactions that would slow down the dynamics.

## Results

### Measuring dynamics of single-stranded nucleic acids

Our approach is illustrated in Fig. 1. Confocal single-molecule FRET measurements of freely diffusing molecules were used to obtain transfer efficiencies, which can be related to the average distance between the FRET donor and acceptor attached to the ends of the oligonucleotides (Fig. 1a). We focused on ssNAs with 19 nucleotides terminally labeled with Alexa Fluor 488 and 594, which at near-physiological ionic strengths of 153 mM yield transfer efficiencies close

to 0.5, where the sensitivity for distance fluctuations is optimal. To probe the influence of sequence composition on chain dynamics, we studied homopolymeric 19-mer ssDNA and ssRNA oligonucleotides, with cytosine ($dC_{19}$, $rC_{19}$), adenine ($dA_{19}$, $rA_{19}$), and thymine or uracil ($dT_{19}$, $rU_{19}$) as nucleobases. Guanine was excluded from our study because of its propensity to form stable quadruplex structures[33]. To explore the impact of chain length on dynamics, we included a double-length 38-mer of deoxythymidine ($dT_{38}$). Additionally, we examined a partially abasic sequence, consisting of 10 thymine bases alternating with nine sites lacking the base ($dT_{19}^{ab}$). This sequence allowed us to assess the effect of the nucleobases on chain dynamics and the influence of base stacking.

Based on the transfer efficiency histograms, we can single out the FRET-active subpopulation and exclude the contribution of donor-only-labeled molecules and any unwanted subpopulations associated with compact structures (Supplementary Fig. 1) in the analysis of distance dynamics (Fig. 1a,b). The widths of the transfer efficiency peaks of the unstructured ssNAs are close to the photon shot noise limit (Fig. 1a and Supplementary Fig. 1), indicating that interdye distance fluctuations are averaged out during the diffusion time of the molecules through the confocal volume of ~1 ms. Time-resolved fluorescence anisotropy measurements indicate high mobility of the fluorophores (Supplementary Fig. 4, Supplementary Table 2), suggesting rapid averaging of the relative orientations of the dyes[34]. However, the fluorescence lifetimes observed for donor and acceptor

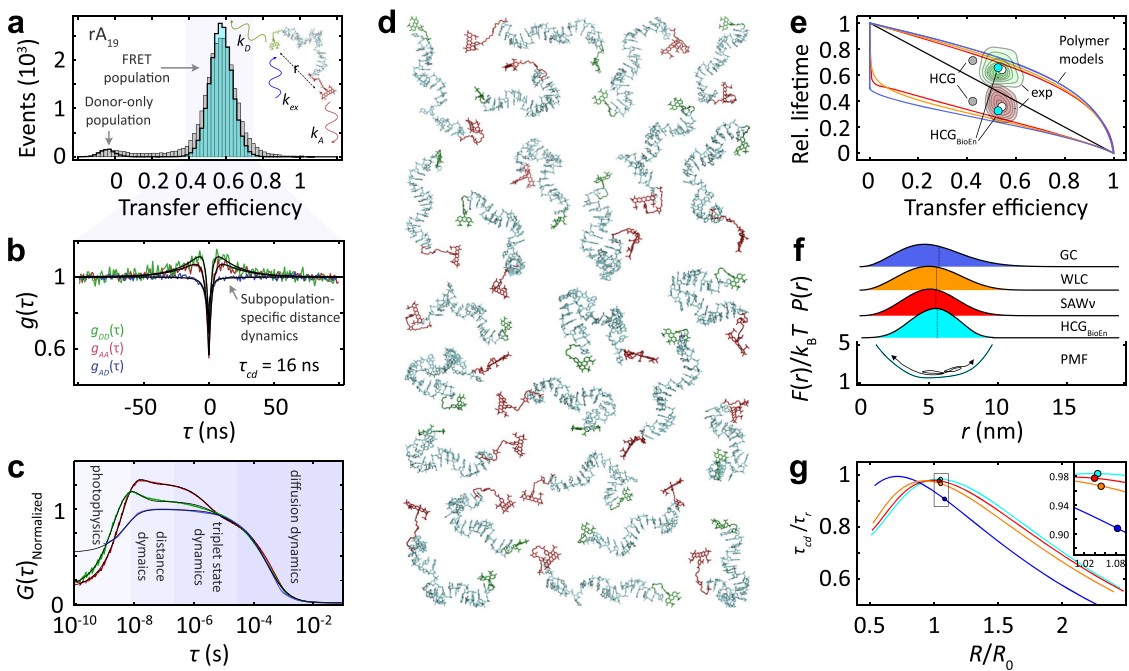

**Fig. 1 | Quantifying chain reconfiguration dynamics of single-stranded nucleic acids (illustrated for rA₁₉). a** Transfer efficiency histogram of freely diffusing terminally labeled $rA_{19}$ at 150 mM NaCl in 10 mM HEPES pH 7, with the FRET-active population at $E \approx 0.55$ and the donor-only population at $E \approx 0$ from molecules with inactive acceptor (gray: measured; black line and cyan shading: shot noise-limited photon distribution analysis[57]) with an inset of a schematic representation of FRET on ssNA. **b** Normalized nsFCS of the FRET subpopulation shaded in panel **a** with donor (green) and acceptor (red) fluorescence autocorrelations and donor-acceptor crosscorrelation (blue; black lines: fits with Eq. 12 (SI Methods), with resulting fluorescence correlation time, $\tau_{cd}$). **c** Normalized subpopulation-specific complete correlation functions (black lines: global fits with Eq. 11). **d** Representative structures from the HCG ensemble of $rA_{19}$ with explicit donor and acceptor dyes, Alexa Fluors 488 (green) and 594 (red). **e** Distributions of relative donor (green contours) and acceptor fluorescence lifetimes (red contours) versus transfer efficiency[8] for all detected bursts (white points: average lifetimes and efficiencies).

The straight line shows the theoretical dependence for fluorophores at a fixed distance (static line); curved lines show the dependences for dynamic systems based on analytical polymer models: Gaussian chain[8] (GC, blue), worm-like chain[5,8] (WLC, orange), modified self-avoiding walk polymer[8,63] (SAW-v, red); upper lines, donor lifetime; lower lines, acceptor lifetime. Gray (HCG) and cyan dots (HCG$_{BioEn}$) show the values from the HCG ensemble and the reweighted ensemble of $rA_{19}$, respectively. **f** Dye-to-dye distance distributions inferred from the mean and variance of the transfer efficiency distributions of $rA_{19}$ for the polymer models and the HCG$_{BioEn}$ ensemble (cyan), with root mean square end-to-end distances, $R$, indicated as vertical lines, and the potential of mean force (PMF) for the HCG$_{BioEn}$ ensemble (SI/Methods). **g** Ratio of fluorescence correlation time ($\tau_{cd}$) and chain reconfiguration time $\tau_r$ (SI Methods) as a function of $R/R_0$, for the different models (circles: values for distance distributions in **f**; $R_0$: Förster radius). Source data are provided as a Source Data file.

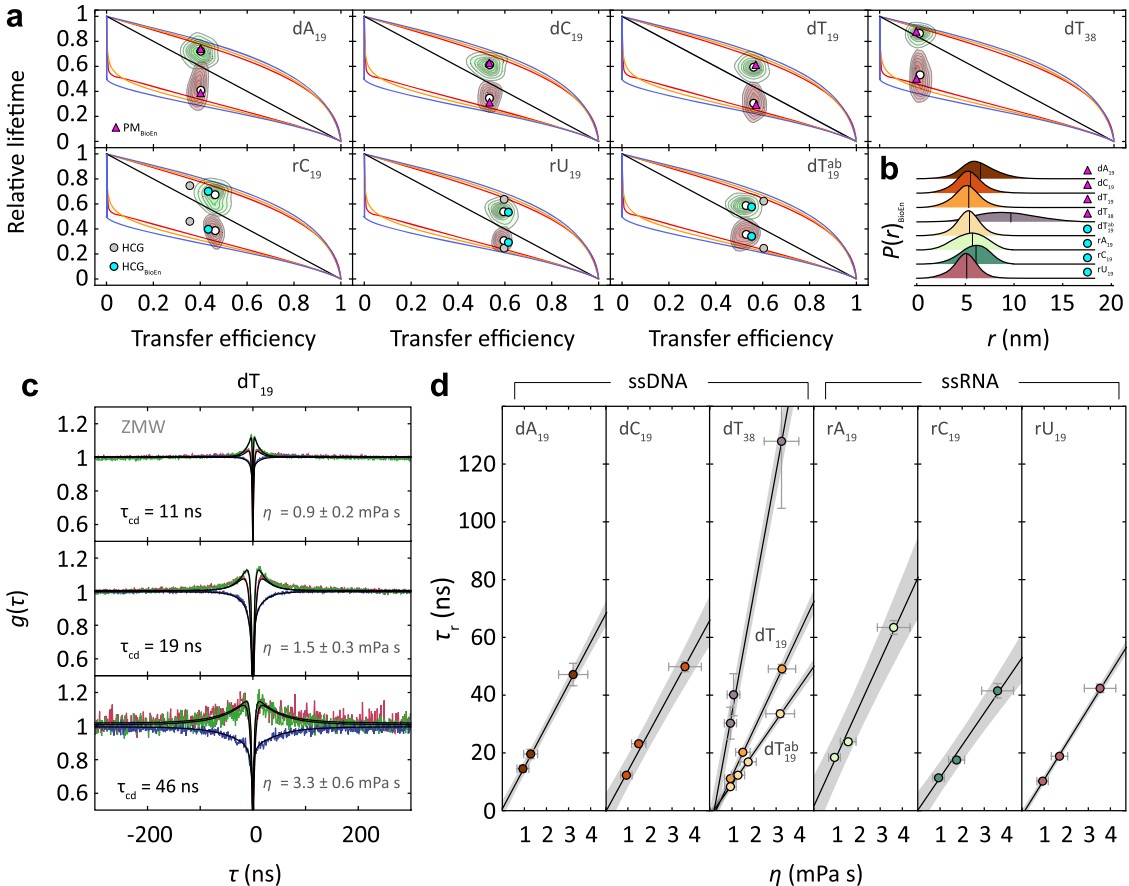

**Fig. 2 | Sequence dependence of chain dynamics and internal friction in single-stranded nucleic acids. a** Distributions of relative donor (green contours) and acceptor fluorescence lifetimes (red contours) versus transfer efficiency of ssDNA and ssRNA from all detected fluorescence bursts (white points: average lifetime and efficiency) compared with predictions from analytical polymer models (analogous to Fig. 1e), the HCG (gray) and HCG$_{BioEn}$ ensembles (cyan) for ssRNA and the reweighted polymer models (PM$_{BioEn}$; purple triangles) for the ssDNA. **b** Dye-to-dye distance distributions from HCG$_{BioEn}$ ensembles (cyan) and reweighted polymer models (purple triangles) with root mean square end-to-end distance, $R$, indicated as vertical lines. **c** Representative normalized subpopulation-specific nsFCS measurements (color code as in Fig. 1) for dT$_{19}$ at different viscosities (uncertainties

correspond to standard deviations of the solvent viscosities estimated from diffusion times using a calibration curve derived from FCS measurements, see Methods) with fluorescence correlation times, $\tau_{cd}$ (black lines, global fits; see Methods). **d** Solvent viscosity ($\eta$) dependence of chain reconfiguration times, $\tau_r$, of ssNAs (color code according to distance distributions as shown in b) with linear fits (shaded bands: 95% confidence intervals). Values and error bars for $\tau_r$ at the viscosity of water are the mean and standard deviation, respectively, from three independent measurements. Values and error bars for $\tau_r$ at higher viscosities represent means and standard deviations from the values obtained based on four different distance distributions (GC, WLC, SAW-ν, HCG$_{BioEn}$). Source data are provided as a Source Data file.

strongly deviate from the values expected for a fixed distance (Figs. 1e, 2a), demonstrating that the interdye distances sample a broad distribution[35,36]. To probe the timescale of the corresponding chain dynamics, we used subpopulation-specific nsFCS[5,8] based on the fluorescence fluctuations of donor and acceptor emission down to the nanosecond range (Fig. 1b).

A characteristic signature of distance dynamics in FRET − in contrast to intensity fluctuations owing to contact quenching or triplet blinking − is that both the donor and acceptor autocorrelations, as well as the donor-acceptor crosscorrelation relax with the same time constant, but with positive correlation amplitudes for the autocorrelations and a negative amplitude for the crosscorrelation[8] (Figs. 1b, 1c, 2c and Supplementary Fig. 2). Global fitting of the three correlation functions (see Methods) indicates that the relaxation corresponding to distance fluctuations is remarkably rapid, with correlation times of $\tau_{cd} \approx 10$ ns (~20 ns for dT$_{38}$). To facilitate measurements of these rapid dynamics, we employed zero-mode waveguides[37] (ZMWs), which speed up data collection for nsFCS by orders of magnitude through fluorescence enhancement[38]. Even more importantly, they lead to reduced fluorescence lifetimes, which improves the time separation between

photon antibunching and distance dynamics in the correlations[38]. On longer timescales, the cross correlations do not exhibit additional components with negative amplitude, indicating the absence of slower distance dynamics (Figs. 1c, 2c), in line with the near shot noise-limited width of the transfer efficiency histograms (Fig. 1a, Supplementary Fig. 1). The decay components of the autocorrelations in the microsecond range are not accompanied by a detectable crosscorrelation component, and are thus most likely to be caused by triplet blinking[38] (Fig. 1c).

**Distance distributions and dynamics**

To obtain from the measured nsFCS relaxation times, $\tau_{cd}$, the chain reconfiguration times, $\tau_r$, i.e., the decorrelation time of the end-to-end distance, we approximate the chain dynamics in terms of diffusion in a potential of mean force derived from the end-to-end distance distribution sampled by the chain[5,8,39] (Fig. 1f and Supplementary Fig. 5). For unfolded and disordered proteins as well as double-stranded DNA, simple analytical models based on polymer theory have been shown to provide suitable approximations of their distance distributions[8,40,41], but for ssNAs, the applicability of such simple models has not been

established. Indeed, obvious candidates for analytical distance distributions, such as the worm-like chain, a self-avoiding walk polymer, or a Gaussian chain, are not in accord with the combined analysis of transfer efficiencies and fluorescence lifetime data (Figs. 1e, 2a, Supplementary Fig. 3). In particular, the deviations from the diagonal static FRET line[42] for these simple polymer models are greater than observed experimentally, which indicates that they overestimate the widths[36] of the distance distributions in ssNAs.

To address this deficiency, we used the recently developed hierarchical chain growth (HCG) approach[31], which has been shown to yield conformational ensembles of single-stranded oligoribonucleotides that are in accord with small-angle X-ray scattering and FRET results[32]. Briefly, HCG creates a pool of short oligonucleotide structures that are then combined at random into polymers by fragment assembly. By structurally aligning the individual fragments and rejecting fragment pairs that are poorly aligned or involve steric clashes, ensembles with a high quality of both local and global structural properties are obtained (Fig. 1d–f). To account for the FRET dyes, a library of dye and linker configurations from molecular dynamics simulations[29] was used. The resulting ensembles thus include explicit representations of the fluorophores and also take into account that the excluded volume of the dyes affects the conformational distributions of the fluorescently labeled nucleic acid chains.

The resulting ensembles, containing 10,000 members each, were reweighted using Bayesian inference of ensembles[43] (BioEn) to reach agreement with both the means and the variances of the transfer efficiency distributions observed experimentally (Figs. 1e, 2a; see SI, Methods). It is worth emphasizing that we use for this approach not the variance of the transfer efficiency histogram (Fig. 1a), which is dominated by shot noise[44]; rather, we use the variance of the transfer efficiency distribution that corresponds to the underlying distance distribution, which can be obtained from the deviations of the mean fluorescence lifetimes from the static FRET line (Figs. 1e, 2a) if the distance dynamics are slower than the fluorescence lifetimes[35,36,42]. Correspondingly, the reweighting takes into account not only experimental information on the average intramolecular distance but also on the variance of the distance distribution. For the ssRNA sequences, the HCG ensembles yield means and variances close to the experimental values even without reweighting, in line with recent results[32]; slight reweighting thus suffices in these cases. For the ssDNA sequences, much stronger reweighting is required to obtain agreement with experiment, which may point to deficiencies in the current force fields available for DNA[29]. For ssDNA, we thus employed reweighting of the distance distributions from simple analytical polymer models using both the transfer efficiency mean and variance observed experimentally (Fig. 2a; see SI, Methods). The resulting distance distributions indicate that the chain dimensions are similar for the different ssNAs (SI, Methods), but some differences are noteworthy. For instance, the most expanded sequence is dA$_{19}$, in accord with the pronounced base stacking expected for adenine[19,45].

The reweighted distance distributions were then converted to potentials of mean force by Boltzmann inversion (Fig. 1f, Supplementary Fig. 5). The fluorescence relaxation times, $\tau_{cd}$, from nsFCS, combined with the reweighted distance distributions, and the known distance dependence of the FRET efficiency according to Förster's theory[46] fully define the dynamics of the chain in the framework of diffusion in a potential of mean force[5,8,39] (SI, Methods); the dynamics can be characterized either by the effective end-to-end diffusion coefficient or the chain reconfiguration time, $\tau_r$ (Supplementary Table 4). The numerical values of $\tau_{cd}$ and $\tau_r$ are very similar, which is expected[39], because the average transfer efficiencies probed here correspond to distances near $R_0$ (Fig. 1g, Supplementary Table 4). To assess the influence of the detailed shape of the potential on the values of $\tau_r$, we compared the results based on the reweighted HCG ensembles and different analytical polymer models (Fig. 1f). The resulting

maximum differences in $\tau_r$ range from 6 to 17 % for the different 19 mers (Supplementary Table 4), indicating that the reconfiguration times we infer are robust, presumably because by FRET with our dye pair, we primarily probe the central regions of the distance distributions, which are similar for all models (Fig. 1f).

The resulting values of $\tau_r$ for the different 19 mers range from 9 to 17 ns, indicating rapid chain dynamics with a similar timescale for all sequences we investigated (Fig. 2d, Supplementary Table 4), even though very different degrees of base stacking are expected for the different nucleotides. Adenine, e.g., is known to exhibit pronounced stacking, whereas thymine shows low stacking propensity[16,45,47]. dT$_{19}$ and dT$_{19}^{ab}$ (where every other nucleotide is lacking the base) have very similar reconfiguration times and average end-to-end distances, indicating the absence of base stacking in dT$_{19}$ under our experimental conditions. The reconfiguration times are comparable for corresponding ssRNA and ssDNA samples, suggesting that the dynamics are dominated by the four freely rotatable bonds in the phosphodiester linkage between nucleotides rather than the identity of the nucleobase or sugar. Nevertheless, more pronounced stacking appears to be correlated with somewhat slower end-to-end distance dynamics, as observed previously in quenching experiments[23].

## Role of internal friction

To obtain more mechanistic insight into ssNA dynamics, we quantified the contribution of internal friction. Internal friction in biomolecules describes the dissipative force resisting conformational changes or molecular motion that is not caused by friction against the solvent but by the motion of parts of the molecules with respect to each other[48–50]. A commonly used operational definition of internal friction is based on measurements of the dynamics as a function of solvent viscosity, and the frictional contribution independent of the solvent is obtained by extrapolating to zero viscosity[24,50]. This concept is particularly well justified for polymers, where, in the context of the Rouse and Zimm models of chain dynamics with internal friction[50–52], the total reconfiguration time of the chain, $\tau_r$, can be decomposed into two additive terms, the contribution from internal friction, $\tau_i$, which is independent of solvent viscosity, $\eta$, and the solvent viscosity-dependent term, $\tau_s$:

$$\tau_r \approx \tau_i + \frac{\eta}{\eta_o}\tau_s(\eta_o) \qquad (1)$$

where $\eta_0$ is the viscosity of water. $\tau_i$ thus corresponds to the value of $\tau_r$ extrapolated to $\eta = 0$ (Fig. 2d).

To quantify internal friction, we varied the solvent viscosity by changing the glycerol concentration (Fig. 2d, Supplementary Fig. 2). Strikingly, the resulting values of $\tau_i$ are zero within experimental uncertainty for all ssNAs investigated (Fig. 2d), suggesting that internal friction makes a negligible contribution to their chain dynamics. In sequences with little or no base stacking, such as dT$_{19}$, this observation may not be surprising and is reminiscent of highly expanded unfolded and disordered proteins with little intrachain interactions that could slow down the dynamics[8], but it may be more surprising for sequences with pronounced base stacking, such as d/rA$_{19}$. A possible interpretation that emerges from the dominant configurations from the HCG approach (Fig. 1) and previous simulations[29] is that the end-to-end distance dynamics in such sequences are dominated by the rapid motions of relatively long stacked segments rotating about a few nucleotides where stacking is absent. The motion of those stacked segments through the solvent would hardly be impeded by interactions within the chains and would be dominated by solvent friction.

## Discussion

In contrast to our results, Uzawa et al.[23,24] observed a small but significant contribution of internal friction for the rates of end-to-end contact formation in single-stranded oligodeoxynucleotides with

lengths similar to those used here. However, the two experiments probe very different parts of the end-to-end distance distribution: While FRET with the Förster radii used here is dominated by distance fluctuations about the center of the distributions, contact quenching probes their short-distance tails. It is conceivable that forming contacts between the chain termini requires conformational rearrangements that involve more pronounced barriers, corresponding to higher internal friction. With the distance distributions and effective diffusion coefficients from our results, we estimate end-to-end collision rates[26,39] (see Methods) between $0.2 \times 10^6 \, s^{-1}$ and $2 \times 10^6 \, s^{-1}$ for $dT_{19}$, depending on the distance distribution used (Supplementary Table 5). Despite the pronounced dependence on the detailed shape of the distribution, the contact rate of $1.25 \times 10^6$ reported by Uzawa et al.[23] for $dT_{20}$ is within this range.

In summary, with single-molecule FRET and nsFCS aided by nanophotonic enhancement, we observe very rapid chain dynamics of ssNAs. In combination with conformational ensembles generated by hierarchical chain growth, simple polymer models, and reweighting based on experimental restraints, we obtain reconfiguration times in the 10-ns range, and we observe no detectable contribution of internal friction. These dynamics are much faster than for most unfolded or disordered proteins with similar average end-to-end distances. Especially for unfolded proteins under native solution conditions, reconfiguration times up to several hundred nanoseconds have been observed[8,26], often with a pronounced contribution from internal friction dominated by intrachain interactions[26,53]. The more rapid reconfiguration in ssNAs may be linked to their larger persistence lengths (Supplementary Table 3) and the hinge-like motions of partially stacked segments relative to each other, which are expected to be dominated by solvent friction. It will be interesting to relate the fast dynamics of single-stranded nucleic acids to processes such as structure formation and binding.

## Methods

### Purification and labeling of nucleic acids

Terminally functionalized homopolymeric oligonucleotides with a 5′-end dithiol and a 3′-end primary amine for labeling ($dA_{19}$, $dC_{19}$, $dT_{19}$, $dT_{19}^{ab}$, $dT_{38}$, $rA_{19}$, $rC_{19}$ and $rU_{19}$; Supplementary Table 1) were synthesized and purified by high pressure liquid chromatography (HPLC) by Integrated DNA Technologies. Prior to labeling, the oligonucleotides were dissolved in 10 mM sodium phosphate buffer pH 7, and filtered and concentrated (Amicon Ultra-0.5 mL, MWCO 3 KDa) to remove free primary amines that interfere with downstream reactions. After this step, each oligonucleotide was site-specifically labeled at the 5′-end with thiol-reactive Alexa Fluor 594 maleimide, and at the 3′-end with amine-reactive Alexa Fluor 488 succinimidyl ester according to the following procedure. The synthetically incorporated thiol groups at the 5′-ends were reduced with 100 mM tris(2-carboxyethyl)phosphine (TCEP) at oligonucleotide concentrations of ~10 μM. After ~1 h, the buffer of the samples was exchanged to 10 mM sodium phosphate pH 7, and the samples were concentrated (Amicon Ultra-0.5 mL Centrifugal Filters MWCO 3 KDa) to ~10 μM. The acceptor dye (dissolved in 5 μL dimethyl sulphoxide (DMSO) and vortexed) was added to the sample at a ratio of 10:1 (dye:oligonucleotide) and incubated for 60 min. For the reaction of the amine-reactive donor dye with the 3′-end of the oligonucleotides, the pH of the acceptor labeling reaction mixtures was increased to pH 8 by addition of 1 M sodium phosphate buffer pH 8. The donor dye (dissolved in 5 μl DMSO and vortexed) was added to the corresponding reaction mix in a tenfold excess over oligonucleotide and incubated for 60 minutes. Unreacted dye was removed with a desalting spin column (Zeba, Pierce, MWCO 7 kDa), and the labeled constructs were purified on a reversed-phase column (Dr. Maisch ReprosilPur 200 C18-AQ, 5 μm) using HPLC (Agilent 1100 series). Samples were lyophilized overnight, then dissolved in $H_2O$, and stored at −80 °C until use.

### Production of ZMWs

Using borosilicate glass coverslips coated with a 100 nm aluminum layer (Deposition Research Laboratory, St. Charles, MO), ZMWs with a diameter of 120 nm were milled into the aluminum layer at a 90° angle using a gallium focused ion beam (FIB-SEM Zeiss Auriga 40 Cross-Beam) with a voltage of 30 kV and a 10 pA beam current at room temperature. Before the experiments, the ZMWs were cleaned with double-distilled water and ≥ 99.7% ethanol to remove dust. They were then exposed to a 5 min air plasma treatment, followed by a 12 h incubation at room temperature in nitrogen atmosphere with 1 mg/mL of silane-modified PEG 1000 dissolved in ethanol and 1% acetic acid. After incubation, the ZMWs were washed with ethanol and 1% Tween 20 to remove excess PEG-silane, followed by a final rinse with ethanol and water before air drying.[38,54]

### Single-molecule spectroscopy

Single-molecule fluorescence experiments with and without ZMWs were performed on a four-channel MicroTime 200 confocal instrument (PicoQuant) equipped with either an Olympus UplanApo 60x/1.20 Water objective for measurements without ZMW or an Olympus UplanSapo 100x/1.4 Oil objective for measurements with ZMWs. Alexa 488 was excited with a diode laser (LDH-D-C-485, PicoQuant) at an average power of 100 μW (measured at the back aperture of the objective). The laser was operated in continuous-wave mode for nsFCS experiments and in pulsed mode with interleaved excitation (PIE)[55] for fluorescence lifetime measurements. The wavelength range used for acceptor excitation was selected with two band pass filters (z582/15 and z580/23, Chroma) from the emission of a supercontinuum laser (EXW-12 SuperK Extreme, NKT Photonics) operating at a pulse repetition rate of 20 MHz (45 μW average laser power after the band pass filters). The SYNC output of the SuperK Extreme was used to trigger interleaved pulses from the 488-nm diode laser. Sample fluorescence was collected by the microscope objective, separated from scattered light with a triple band pass filter (r405/488/594, Chroma) and focused on a 100-μm pinhole. After the pinhole, fluorescence emission was separated into two channels, either with a polarizing beam splitter for fluorescence lifetime measurements, or with a 50/50 beam splitter for nsFCS measurements to avoid the effects of detector deadtimes and afterpulsing on the correlation functions[5]. Finally, the fluorescence photons were distributed by wavelength into four channels by dichroic mirrors (585DCXR, Chroma), additionally filtered by band pass filters (ET 525/50 M and HQ 650/100, Chroma), and focused onto one of four single-photon avalanche detectors (SPCM-AQRH-14-TR, Excelitas). The arrival times of the detected photons were recorded with a HydraHarp 400 counting module (PicoQuant, Berlin, Germany).

All free-diffusion single-molecule experiments were conducted with labeled oligonucleotide concentrations between 100 and 250 pM without ZMWs or between 50 and 300 nM with ZMWs in 10 mM HEPES buffer pH 7.0 (adjusted with 35 mM NaOH), 0.01% Tween 20, 143 mM β-mercaptoethanol (BME), 150 mM NaCl, and for the viscosity dependence with appropriately chosen concentrations of glycerol (without ZMW) in 18-well plastic slides (ibidi) or in ZMWs at 22 °C.

### Single-molecule FRET data analysis

Data analysis was carried out using the Mathematica (Wolfram Research) package Fretica (https://github.com/SchulerLab). For the identification of photon bursts, the photon recordings were time-binned (1 ms binning for measurements without ZMWs, 0.2 ms for measurements with ZMWs). Photon numbers per bin were corrected for background, crosstalk, differences in detection efficiencies and quantum yields of the fluorophores, and for direct excitation of the acceptor[56]. Bins with more than 50 photons were identified as photon bursts. Ratiometric transfer efficiencies were obtained for each burst from $E = n_A/(n_A + n_D)$, where $n_A$ and $n_D$ are the corrected numbers of donor and acceptor photons in the photon burst, respectively. The

$E$ values were histogrammed. The subpopulation corresponding to the FRET-labeled species was fitted with a Gaussian peak function or analyzed by photon distribution analysis taking into account the experimentally observed burst size distribution[57–59] (Supplementary Fig. 1). Bursts from experiments in PIE mode were further selected according to the fluorescence stoichiometry ratio[60–62], $S$ ($0.2 < S < 0.8$) (Fig. 2, Supplementary Fig. 3).

## End-to-end distance distributions

For analyzing the single-molecule FRET data of the ssNA variants, we employed end-to-end distance distributions of analytical polymer models as well as the distance distributions obtained by the hierarchical chain growth (HCG) approach[31] (see hierarchical chain growth). The three polymer models used and the corresponding end-to-end distance probability density functions were:

Gaussian chain (GC)[8]:

$$P_{GC}(r) = 4\pi r^2 \left[ \frac{3}{2\pi\langle r^2\rangle} \right]^{\frac{3}{2}} e^{-\frac{3}{2}\frac{r^2}{\langle r^2\rangle}} \quad (2)$$

Worm-like chain (WLC)[8,41]:

$$P_{WLC}(r) = \begin{cases} C(r/l_c)^2 \left(1 - (r/l_c)^2\right)^{-9/2} e^{-3l_c/\left[4l_p\left(1-(r/l_c)^2\right)\right]}, & r \le l_c \\ 0 & , \quad r > l_c \end{cases} \quad (3)$$

where $l_c$ and $l_p$ are the contour- and persistence lengths of the chain, respectively (Supplementary Table 3). $C$ is a normalization constant.

Self-avoiding walk polymer (SAW-$\nu$)[63]:

$$P_{SAW-\nu}(r) = A\frac{4\pi}{R}\left(\frac{r}{R}\right)^{2+g} e^{-\alpha\left(\frac{r}{R}\right)^{\delta}},$$
$$\text{with } R = \langle r^2\rangle^{\frac{1}{2}}, \ g = \frac{\gamma-1}{\nu}, \ \delta = \frac{1}{1-\nu}, \ \gamma \approx 1.1615, \text{ and } \nu = \frac{\ln(\frac{R}{b})}{\ln(n)}, \quad (4)$$

where $b$ and $n$ are the segment length and the number of segments of the polymer, respectively (Supplementary Table 3). $A$ is a normalization constant.

If the rotational correlation time, $\tau_{rot}$ (Supplementary Table 2), of the chromophores is short relative to the fluorescence lifetime, $\tau_D$, of the donor (such that orientational factor $\kappa^2 \approx 2/3$)[34], and the end-to-end distance dynamics of the polypeptide chain (with relaxation time $\tau_r$) are slow relative to $\tau_D$, the experimentally determined mean transfer efficiency, $\langle E\rangle$, can be related to the distance distribution, $P(r)$, by[64]:

$$\langle E\rangle = \langle\varepsilon\rangle \equiv \int_0^\infty \varepsilon(r)P(r)dr, \quad (5)$$

where $\varepsilon(r) = R_0^6/(R_0^6 + r^6)$, and $R_0$ is the Förster radius (5.4 nm for Alexa 488/594)[26,38,65,66]. See Supplementary Table 3 for the values of the parameters used ($l_c$, $b$, $n$) and inferred ($R$, $l_p$, $\nu$) by solving Eq. 5 numerically for the corresponding variable. Time-resolved fluorescence anisotropy measurements (Supplementary Fig. 4) indicate high mobility of the fluorophores (Supplementary Table 2), suggesting $\kappa^2 \approx 2/3$.

## Effect of glycerol on conformational free energy

Viscogens can affect intramolecular interactions and thus lead to changes in conformational free energy. The changes in transfer efficiency upon addition of glycerol were small under the conditions used here; we estimated the order of magnitude of the effect based on a simple approximation. Distance distributions, $P(r)$, can be converted into potentials of mean force, $F(r)$, through Boltzmann inversion (Fig. 1f, Supplementary Fig. 5):

$$F(r) = -k_B T \ln P(r), \quad (6)$$

where $k_B$ is the Boltzmann constant and $T$ the temperature. To estimate the change in conformational free energy upon addition of glycerol, we utilized Eq. 6 for the ssNA that exhibited the largest influence of glycerol on transfer efficiency, $dT_{19}$ ($\Delta E = 0.07$), assuming a Gaussian chain distance distribution, with $R_0$ corrected for the refractive index change due to glycerol. The free energy change was estimated from

$$\frac{\Delta F}{k_B T} = \int_o^\infty P_{GC}^{(35\%)}(r)\ln P_{GC}^{(35\%)}(r)\,dr - \int_o^\infty P_{GC}^{(0\%)}(r)\ln P_{GC}^{(0\%)}(r)\,dr = \ln\sqrt{\frac{\langle r_{35\%}^2\rangle}{\langle r_{0\%}^2\rangle}}, \quad (7)$$

where $P_{GC}^{(0\%)}(r)$ and $P_{GC}^{(35\%)}(r)$ are the probability density functions of the chain at 0% and 35% glycerol, respectively, and $\langle r_{0\%}^2\rangle$ and $\langle r_{35\%}^2\rangle$ are the corresponding mean squared end-to-end distances. The resulting conformational free energy change is $\Delta F \approx 0.1 k_B T$, corresponding to a change in $R = \langle r^2\rangle^{1/2}$ by -0.7 nm. We thus conclude that the energetic changes within the chain upon glycerol addition are unlikely to affect internal friction.

## Single-molecule fluorescence lifetime analysis

From PIE experiments, the donor and acceptor fluorescence lifetimes, $\tau_D$ and $\tau_A$, for each burst were determined from the mean detection times, $\tau'_D$ and $\tau'_A$, of all photons of a burst detected in the donor and acceptor channels. These times are measured relative to the preceding pulses of the laser triggering electronics. Photons of orthogonal polarization with respect to the excitation polarization were weighted by $2\,G$ to correct for fluorescence anisotropy effects; $G$ corrects for the polarization-dependence of the detection efficiencies. For obtaining the mean fluorescence lifetimes, we further corrected for the effect of background photons and for a time shift due to the instrument response function (IRF) with $\tau_{x=D,A} = \frac{\tau'_x - \alpha\langle t\rangle_{bg,x}}{1-\alpha} - \langle t\rangle_{IRF}$, with $\alpha = n_{bg,x}\Delta/N_x$. Here, $\langle t\rangle_{bg,x}$ is the mean arrival time of the background photons, $\langle t\rangle_{IRF}$ is the mean time of the IRF, $n_{bg,x}$ is the background photon detection rate, $\Delta$ the burst duration, and $N_x$ the uncorrected number of photons in the donor ($x = D$) or acceptor ($x = A$) channels[67]. The distributions of relative lifetimes, $\tau_D/\tau_{D0}$ and $(\tau_A - \tau_{A0})/\tau_{D0}$, versus transfer efficiency for the FRET-active population are shown in Figs. 1e, 2a and Supplementary Fig. 3. $\tau_{D0}$ and $\tau_{A0}$ are the fluorescence lifetimes of donor and acceptor in the absence of FRET, respectively (see Supplementary Table 2). Supplementary Fig. 3 shows the distributions of relative donor lifetime versus transfer efficiency including the donor-only population. $\tau_{A0}$ and $\tau_{D0}$ were obtained from independent ensemble lifetime measurements as described below. The figures show dynamic FRET lines[42] that were calculated assuming end-to-end distance distributions, $P(r)$, for a Gaussian chain[8] (GC), a worm-like chain[8] (WLC), and for the SAW-$\nu$ polymer[63] (SAW-$\nu$) models. For the case that $P(r)$ is sampled faster than the interphoton time (~10 μs) but slowly compared to $\tau_D$ (3.5–4 ns; Supplementary Table 2), it has been shown that[36]

$$\frac{\tau_D}{\tau_{D0}} = 1 - \langle\varepsilon\rangle + \frac{\sigma_\varepsilon^2}{1 - \langle\varepsilon\rangle},$$

and

$$\frac{\tau_A - \tau_{A0}}{\tau_{D0}} = 1 - \langle\varepsilon\rangle - \frac{\sigma_\varepsilon^2}{\langle\varepsilon\rangle}, \quad (8)$$

where $\sigma_\varepsilon^2 = \int_0^\infty (\varepsilon(r) - \langle\varepsilon\rangle)^2 P(r)dr$ is the variance of the transfer efficiency distribution corresponding to $P(r)$. The dynamic FRET lines

were obtained by varying the model parameters of the corresponding distributions, $\langle r^2 \rangle$ for the GC; the persistence length, $l_p$, for the WLC; and the scaling exponent, $\nu$, for the SAW-$\nu$ model, respectively. The static FRET line, $\tau_D/\tau_{D0} = (\tau_A - \tau_{A0})/\tau_{D0} = 1 - \langle \varepsilon \rangle$, corresponds to fixed interdye distances. Note that this type of fluorescence lifetime analysis is only valid for the regime where $\tau_{rot}$ is short relative to $\tau_D$, and $\tau_D$ is short relative to $\tau_r$, i.e., $\tau_r < \tau_D < \tau_r$ (Supplementary Table 2, Supplementary Fig. 4).

## Hierarchical chain growth and fluorophore modeling

To carry out hierarchical chain growth (HCG), we created a molecular dynamics (MD) fragment library. Subsequently, we built heterotetramers with sequence d/rGXYZ. G served as a fixed head group at the 5′ end. For the other nucleotides "XYZ", we used all $4^3$ combinations of thymine, uracil, cytosine and adenine. The heteromeric fragment library was extensively sampled via temperature replica exchange MD simulations, utilizing the parmBSC1 force-field[68] for DNA and the DESRES force-field[30] for RNA. For both DNA and RNA, the TIP4P-D water model[69] was used. Fragments were placed in a dodecahedral box, solvated with 150 mM NaCl and neutralized, resulting in a system comprising ~6600 atoms. Depending on the fragment sequence, the total number varied by about 50 atoms. The fragment with the abasic site was parameterized as described by Heinz *et al.*[70]. MD simulations were performed using GROMACS/ 2019.6.[71] For each system, we ran 24 replicas over a temperature range of 300–420 K for 100 ns as described before[32]. Afterwards, we randomly selected fragment conformations from the MD fragment library at 300 K to assemble disordered ssNAs with HCG in a hierarchical manner[32].

We also used HCG to build libraries of dye-labeled DNA and RNA 4-mer fragments. As inputs, we used the 4-mer libraries built here and the libraries built by Grotz et al.[29] for the dyes Alexa Fluor 594 and Alexa Fluor 488 attached to dideoxyadenosine monophosphate ($dA_2$) and dideoxythymidine monophosphate ($dT_2$) at the 5′ and 3′ ends, respectively. The use of $dA_2$- and $dT_2$-dye fragments to model fluorophores attached to DNA and RNA chains has been validated by Grotz et al.[29] We used the $dA_2$ library for purines (A, G) at the respective end and the $dT_2$ library for pyrimidines (U, C). Pairs of random structures were repeatedly drawn from the library of DNA or RNA 4-mer fragments and from library of $dA_2$ or $dT_2$ labeled with Alexa 594 or Alexa 488. For each pair, we performed a rigid body superposition of the heavy atoms of the terminal sugar moiety and nucleobase, leaving out non-matching atoms of the base. If the RMSD of the superposition was below 0.8 Å, we searched for clashing heavy atoms within a pair distance of 2.0 Å. If no clashing atoms were detected, the dye was attached to the DNA or RNA 4-mer fragment according to the superposition, excluding the terminal oxygen atoms of the 4-mer. The resulting libraries of DNA and RNA 4-mer structures with the FRET dyes Alexa Fluor 594 and Alexa Fluor 488 attached at their 5′ or 3′ ends were subsequently used to build dye-labeled DNA and RNA chains by HCG.

## Bayesian ensemble refinement

To optimize the agreement of the HCG ensembles with the experimental data, we reweighted the ensembles of configurations based on two experimental observables from the single-molecule measurements: the mean transfer efficiency, $\langle E \rangle$, and the variance of the underlying transfer efficiency distribution, $\sigma_\varepsilon^2$, as described in *Single-molecule fluorescence lifetime analysis* (for experimental $\langle E \rangle$ and $\sigma_\varepsilon^2$ of the individual constructs, see Supplementary Table 6). The transfer efficiency was calculated for each of the $N$ ensemble members, and uniform weights $w_\alpha^0 = 1/N$ were initially assigned to all of them. Optimal weights were found using Bayesian inference of ensembles[43]

(BioEn) by minimizing

$$\Delta G(w_1, \ldots, w_N) = \tfrac{1}{2}\chi^2 - \theta \Delta S$$

$$\text{with } \chi^2 = \frac{\left( \langle \varepsilon \rangle_{\text{BioEn}} - \langle E \rangle \right)^2}{Var(\langle E \rangle)} + \frac{(\sigma_{\varepsilon\,\text{BioEn}}^2 - \sigma_\varepsilon^2)^2}{Var(\sigma_\varepsilon^2)} \quad (9)$$

$$\text{and } \Delta S = - \sum_{\alpha=1}^{N} w_\alpha \ln \frac{w_\alpha}{w_\alpha^0}$$

The optimal weights $w_\alpha$ of ensemble members $\alpha$ are written in terms of two generalized forces[43] $f$ and $g$ for the first and second power of the respective transfer efficiency $\varepsilon_\alpha$, i.e., $w_\alpha \propto w_\alpha^0 \exp(f \varepsilon_\alpha + g \varepsilon_\alpha^2)$ with $\sum_{\alpha=1}^{N} w_\alpha = 1$. The minimum of $\Delta G$ as function of $f$ and $g$ was found with a 2D Newton–Raphson solver, staying in the convex region by first increasing and then step-wise decreasing $\theta$ to the target value. Reweighted values are given by $\langle \varepsilon \rangle_{\text{BioEn}} = \sum_{\alpha=1}^{N} w_\alpha \varepsilon_\alpha$ and $\sigma_{\varepsilon\,\text{BioEn}}^2 = \sum_{\alpha=1}^{N} w_\alpha \varepsilon_\alpha^2 - \langle \varepsilon \rangle_{\text{BioEn}}^2$, where $w_\alpha$ and $\varepsilon_\alpha$ are the weight and the transfer efficiency of the $\alpha$ th ensemble member, respectively, with $\sum_{\alpha=1}^{N} w_\alpha = 1$. For each ensemble, we chose the largest value of $\theta$ for which $\langle \varepsilon \rangle_{\text{BioEn}}$ and $\sigma_{\varepsilon\,\text{BioEn}}^2$ agreed with the measured values within the experimental uncertainties of $Var(\langle E \rangle)^{1/2} = 0.03$ and $Var(\sigma_\varepsilon^2)^{1/2} = 0.003$, respectively. The dye-to-dye distance distributions of initial and reweighted ensembles for all ssRNAs and $dT_{19}^{ab}$ are depicted in Supplementary Fig. 5. A useful quantity to estimate the quality of the prior distribution is the effective fraction of configurations used from the initial ensemble, $\phi_{\text{eff}} = e^{\Delta S}$. For the 19 mer ssRNA ensembles and $dT_{19}^{ab}$, $\phi_{\text{eff}}$ was between 75% and 90%, for the 19 mer ssDNA ensembles, $\phi_{\text{eff}}$ was between 65% and 71%, indicating that the prior distributions for ssRNA were in better agreement with the experimental data than for ssDNA. In view of the strong HCG ensemble reweighting required for ssDNA, which in the case of $dT_{38}$ with its very low transfer efficiency resulted in a bimodal end-to-end distance distribution, we instead reweighted the transfer efficiency distributions obtained from the analytical polymer models for ssDNA. To achieve this, we discretize the transfer efficiency range uniformly between 0 and 1, $\varepsilon_\alpha = (\alpha - 1)\Delta\varepsilon$, with $\Delta\varepsilon = 0.03$, and $\alpha$ ranging from 1 to $N = 33$. We used as priors the distance distributions, $P(r)$, from the analytical polymer models (Eqs. 2–4) to obtain the initial weights $w_\alpha^0 = a P(r(\varepsilon_\alpha)) |\frac{dr}{d\varepsilon_\alpha}|$, where $r(\varepsilon_\alpha) = R_0 (1/\varepsilon_\alpha - 1)^{1/6}$ and $a$ is a normalization constant ensuring $\sum_{\alpha=1}^{N} w_\alpha^0 = 1$. We then minimized $\Delta G$ with respect to $w_1, \ldots, w_N$ as described above. For all three different prior distributions, we found very similar reweighted distributions (i.e., values of $w_\alpha$). The end-to-end distance distributions of prior and reweighted polymer models for all ssDNAs are depicted in Supplementary Fig. 5. We used the reweighted HCG distributions (HCG$_{\text{BioEn}}$) for ssRNA and the reweighted polymer model distributions (PM$_{\text{BioEn}}$) for ssDNA to convert $\tau_{cd}$ to $\tau_r$ (see below, Eqs. 14, 15, Fig. 2, Supplementary Table 4). We note, however, that even with the reweighted distance distributions from HCG for ssDNA, the resulting values of $\tau_r$ are very similar to those from the alternative analyses (Supplementary Table 4).

## Fluorescence correlation spectroscopy (FCS)

FCS measurements were performed on freely diffusing Alexa 488- and Alexa 595-labeled oligonucleotides at concentrations and buffer conditions as described in "Free diffusion single-molecule spectroscopy". Additionally, we included appropriate concentrations of glycerol to increase the solvent viscosity (measurements performed without ZMWs). The correlation between two time-dependent signal intensities, $I_i(t)$ and $I_j(t)$, measured on two detectors $i$ and $j$, is defined as:

$$G_{ij}(\tau) = \frac{\langle I_i(t) I_j(t+\tau) \rangle}{\langle I_i(t) \rangle \langle I_j(t) \rangle} - 1, \quad (10)$$

where the pointed brackets indicate averaging over $t$. In our experiments, we use two acceptor and two donor detection channels, resulting in the autocorrelations $G_{AA}(\tau)$ and $G_{DD}(\tau)$, and cross correlations $G_{AD}(\tau)$ and $G_{DA}(\tau)$. By correlating detector pairs, and not the signal from a detector with itself, contributions to the correlations from deadtimes and afterpulsing of the detectors are eliminated[5,72]. Full FCS curves with logarithmically spaced lag times ranging from nanoseconds to seconds (Fig. 1c) were fitted with[73,74]

$$G_{ij}(\tau) = a_{ij} \frac{\left(1 - c_{ab}^{ij} e^{-\frac{|\tau|}{\tau_{ab}^{ij}}}\right)\left(1 + c_{cd}^{ij} e^{-\frac{|\tau|}{\tau_{cd}}}\right)\left(1 + c_T^{ij} e^{-\frac{|\tau|}{\tau_T^{ij}}}\right)}{\left(1 + \frac{|\tau|}{\tau_D^{ij}}\right)\left(1 + \frac{|\tau|}{s^2 \tau_D^{ij}}\right)^{1/2}} \quad (11)$$

The three terms in the numerator with amplitudes $c_{ab}$, $c_{cd}$, $c_T$ and timescales $\tau_{ab}, \tau_{cd}, \tau_T$ describe photon antibunching, chain dynamics, and triplet blinking, respectively. $\tau_D$ is the translational diffusion time of the labeled molecules through the confocal volume; a point spread function (PSF) of 3-dimensional Gaussian shape is assumed, with a ratio of axial over lateral radii of $s = \omega_z/\omega_{xy}$ ($s = 5.3$ without and $s = 1.0$ with ZMW; note that this PSF is not expected to be a good approximation for the confocal volume in the ZMWs but has been commonly used owing to a lack of suitable alternatives[54,75]), and $a_{ij}$ are the amplitudes of the correlation functions. Parameters without indices $ij$ are treated as shared parameters in the global fits of the auto- and crosscorrelation functions. To study the dynamics in more detail, donor and acceptor fluorescence auto- and crosscorrelation curves were computed and analyzed over a linearly spaced range of lag times, $\tau$, up to a maximum, $\tau_{max}$, that exceeds $\tau_{cd}$ by an order of magnitude (Supplementary Fig. 2). For the subpopulation-specific analysis, we used only photons of bursts with $E$ in the range of $\pm 0.2$ of the mean transfer efficiency of the FRET-active population, which reduces the contribution of donor-only and acceptor-only signal to the correlation. For direct comparison, correlation curves were normalized to unity at $\tau_{max}$. After normalization and in the limit of $|\tau| \ll \tau_T^{ij}$ and $|\tau| \ll \tau_D^{ij}$, Eq. 11 reduces to:

$$g_{ij}(\tau) = b_{ij}\left(1 - c_{ab}^{ij} e^{-\frac{|\tau|}{\tau_{ab}^{ij}}}\right)\left(1 + c_{cd}^{ij} e^{-\frac{|\tau|}{\tau_{cd}}}\right), \quad (12)$$

where $b_{ij} = 1/G_{ij}(\tau_{max})$ are the normalization constants.

For quantifying the solvent viscosity directly in the samples as a function of glycerol concentration, we used the information available from the FCS measurements. The average diffusion time of the labeled oligonucleotides through the confocal volume is directly proportional to the solvent viscosity, $\eta$, so $\eta$ can be estimated from an FCS-based calibration curve. Calibration curves were obtained by measuring the diffusion time by means of acceptor and donor autocorrelations and acceptor-donor cross correlations of double-labeled $dT_{19}^{ab}$ at five different known solvent viscosities adjusted with glycerol. The viscosity of each solution was determined using a cone/plate viscometer (DV-I+, Brookfield Engineering Laboratories, Middleboro, MA, USA). Diffusion times were normalized to the diffusion time in buffer and their dependence on viscosity fitted linearly. The solvent viscosity of all other solutions was obtained based on this calibration from the diffusion times of the samples. The values and uncertainties plotted in Fig. 2d represent the resulting means and standard deviations.

## Fluorescence lifetime measurements

To determine the relevant timescales for fluorescence lifetime analysis, we performed polarization-resolved ensemble lifetime measurements of all ssNAs on a custom-built fluorescence lifetime spectrometer[74], which allowed us to determine the fluorescence

lifetimes of Alexa Fluor 488 and 594 as well as the fluorescence anisotropy decays of the dyes conjugated to the different ssNAs. Fluorescence decays of the donor fluorophore were measured on constructs labeled only with Alexa 488. The acceptor fluorescence lifetime decays and corresponding anisotropy decays were measured upon acceptor excitation of double-labeled constructs. All measurements were performed at 150 mM NaCl, 0.01% Tween 20, 0.143 mM BME in 10 mM HEPES buffer with sample concentrations of 50–200 nM. Alexa 488 was excited by a picosecond diode laser (LDH DC 485) at 488 nm with a pulse repetition rate of 40 MHz. Alexa 594 was excited by a supercontinuum light source (SC450-4, Fianium, Southampton, UK), with the wavelength selected using a z582/15 band pass filter and a pulse frequency of 40 MHz. The emitted donor fluorescence was filtered with an ET 525/50 filter (Chroma Technology), and the acceptor fluorescence with an HQ 650/100 filter (Chroma Technology). The emitted photons were detected with a microchannel plate photomultiplier tube (R3809U-50; Hamamatsu City, Japan), and the arrival times were recorded with a PicoHarp 300 photon-counting module (PicoQuant). Intensity decays, $I_{VH}(t)$ and $I_{VV}(t)$, with horizontal and vertical polarizer orientation, respectively, were measured with vertically polarized excitation (Supplementary Fig. 4). The decays were fitted globally with

$$\begin{aligned} I_{VH}(t) &= \beta\left[1 - r_0\left[\alpha e^{-t/\tau_{rot}} + (1-\alpha)e^{-t/\tau_M}\right]\right]e^{-t/\tau_{fl}} + c_{VH} \\ I_{VV}(t) &= G\beta\left[1 + 2r_0\left[\alpha e^{-t/\tau_{rot}} + (1-\alpha)e^{-t/\tau_M}\right]\right]e^{-t/\tau_{fl}} + c_{VV}, \end{aligned} \quad (13)$$

convolved with the instrument response function (IRF, measured with scattered light). $r_0 = 0.38$ is the limiting anisotropy of the dyes[76]; $G$ accounts for the different detection efficiencies of vertically and horizontally polarized light and was obtained for the donor and acceptor intensities from the ratio of the vertical and horizontal emission after horizontal excitation, $G = I_{HV}/I_{HH}$. The offsets $c_{VV}$ and $c_{VH}$ account for background signal. The two rotational correlation times, $\tau_{rot}$ and $\tau_M$, account for fast fluorophore rotation and slower tumbling of the entire labeled molecule, respectively. $\alpha$ represents the fractional amplitude of the fast component; $\beta$ and $\tau_{fl}$ represent the amplitude and relaxation time of the total fluorescence intensity decay, respectively (Supplementary Table 2).

## Chain reconfiguration time $\tau_r$

For any distance-dependent observable, $f(r)$, the correlation time, $\tau_f$, is defined as

$$\tau_f \equiv \int_0^\infty \frac{\langle \delta f(r(t)) \delta f(r(0)) \rangle_r}{\langle \delta f(r)^2 \rangle_r} dt, \quad (14)$$

where $\delta f(r) = f(r) - \langle f(r) \rangle_r$, and $\langle \cdot \rangle_r$ denotes $\langle \cdot \rangle_r = \int \cdot P(r) dr$. The numerator is defined using the joint probability, $P(r_0, r_t)$, of populating at an arbitrary time zero the distance $r_0$ and at a later time $t$ the distance $r_t$. With these definitions, we have $\langle \delta f(r(t)) \delta f(r(0)) \rangle_r = \iint \delta f(r_t) \delta f(r_0) P(r_0, r_t) dr_0 dr_t$. If the dynamics of $r(t)$ are well described as diffusive motion in a potential of mean force, $F(r) = -k_B T \ln P(r)$, then $\tau_f$ can be calculated from[39]

$$\tau_f = \frac{\int_0^\infty P(r)^{-1}\left[\int_0^r \delta f(\rho)P(\rho)d\rho\right]^2 dr}{D \int_0^\infty \delta f(\rho)^2 P(r) dr}, \quad (15)$$

where $D$ is the effective end-to-end diffusion coefficient. From fitting the nsFCS curves, we get the intensity correlation time, $\tau_{cd} = \tau_\epsilon$, where $f(r) = \epsilon(r)$ is the transfer efficiency. We can use Eq. 15 to convert $\tau_{cd}$ to the physically more interesting chain reconfiguration time, $\tau_r$, where $f(r) = r$. We calculated conversion ratios $\theta = \tau_{cd}/\tau_r$ for all distance distributions used. $\theta$ as a function of $R/R_0$ was calculated for the GC, WLC, and the SAW-$\nu$ polymer models, as well as for the HCG$_{BioEn}$

ensembles by varying $R_0$ (Fig. 1g). Note that $\theta$ is independent of $D$. The resulting values of $\tau_r$ are given in Supplementary Table 4.

**End-to-end contact rates**

For comparing end-to-end distance dynamics measured here with published values of end-to-end contact formation rates[23], we used $D$ obtained using Eq. 15 for all polymer models and all ssNA variants to estimate end-to-end contact rates, $k_{ee}$ (see Supplementary Table 5), according to[77]

$$\frac{1}{k_{ee}} = \frac{1}{k_R} + \frac{1}{D}\int_a^\infty \frac{1}{P(r)}\left[\int_r^\infty P(\rho)d\rho\right]^2 dr, \quad (16)$$

where $k_R = qP(a)$ is the reaction-limited rate, with a quenching rate upon contact of $q = 10^{12} s^{-1}$ and a quenching distance of $a = 0.4$ nm.[3,26]

**Reporting summary**

Further information on research design is available in the Nature Portfolio Reporting Summary linked to this article.

## Data availability

The data generated in this study are provided in the Supplementary Information and Source Data file. The DNA and RNA ensembles, the fragment libraries to grow dTab19, dA19, dC19, dT19 with HCG, and the custom force-field parameters for the fragment with the abasic site are available at https://zenodo.org/records/12154848. Source data are provided with this paper.

## Code availability

Fretica, a custom add-on package for Mathematica (Wolfram Research) was used for the analysis of single-molecule fluorescence data and is available at https://github.com/SchulerLab. The code for hierarchical chain growth is available at the GitHub repository https://github.com/bio-phys/hierarchical-chain-growth/. The Fortran code to perform the BioEn reweighting is available at Zenodo (https://zenodo.org/records/12154848).

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

## Acknowledgements
We thank Jérôme Wenger for advice on ZMW fabrication, Gianluca Galeno for assistance with purification and labeling, José María Mateos Melero for assistance with focus ion beam milling, Dmitrii Makarov for discussion and helpful comments on the manuscript, and the Center for Microscopy and Image Analysis at the University of Zurich for access to instrumentation. This work was supported by the Swiss National Science Foundation (B.S., grant IDs 219629, 188695), the University of Kansas (E.D.H.), and the Max Planck Society (G.H.).

## Author contributions
M.N., L.P., E.D.H., G.H., and B.S. designed research; D.N. developed data analysis tools and instrumentation; M.N. performed measurements; L.P. performed and analyzed HCG; M.N., E.D.H., V.v.R., G.H., and D.N. analyzed measurements and simulations; M.N. and R.K.T. fabricated ZMWs; E.D.H., O.M., G.H., and B.S. supervised research; M.N., D.N. and B.S. prepared the manuscript with the help of all authors.

## Competing interests
The authors declare no competing interests.
