## [Peer Review File · Nature Communications]

Nanosecond chain dynamics of single-stranded nucleic acidsREVIEWER COMMENTS

Reviewer #1 (Remarks to the Author):

The manuscript titled "Nanosecond chain dynamics of single-stranded nucleic acids" by Mark F. Nüesch, Lisa Pietrek, Erik D. Holmstrom, and colleagues presents a detailed study on the dynamics of single-stranded DNA (ssDNA) and RNA (ssRNA) oligonucleotides. This work combines single-molecule Förster resonance energy transfer (FRET), nanosecond fluorescence correlation spectroscopy (nsFCS), and nanophotonic (zero mode waveguide) enhancement to elucidate the rapid conformational dynamics and chain reconfiguration times of short ssNAs.

The authors aim to overcome the limitations in resolving the elementary chain dynamics of ssNAs, which are crucial for understanding their folding and function. By employing a sophisticated experimental setup, they manage to quantify the exceedingly rapid chain dynamics in the 10-ns range. Their findings suggest that these dynamics are predominantly influenced by solvent friction, with negligible internal friction, thereby providing a significant insight into the biophysical behavior of ssNAs.

The methodology employed in this study is highly advanced and innovative, combining cutting-edge techniques to probe the dynamics of ssNAs with unprecedented detail. The use of hierarchical chain growth and Bayesian inference for the analysis further strengthens the validity of their conclusions.

The results indicating the lack of detectable internal friction in ssNAs, regardless of the sequence composition, provide a new understanding of ssNA behavior in solution. This finding contrasts with the expected behavior based on the known structure and properties of ssNAs, offering a novel perspective on their dynamic properties.

In summary, the manuscript significantly advances our understanding of ssNA dynamics. It challenges existing notions about internal friction in biomolecules and provides a foundation for further research into the biophysical properties of ssNAs. The manuscript is written exceedingly clearly and present all the experimental and theoretical details required for justifying the made conclusions. I strongly recommend publication as is.

Reviewer #2 (Remarks to the Author):

Nüesch et al. present experimental and computational results that indicate short single-stranded nucleic acid oligomers exhibit rapid chain dynamics with little internal friction, suggesting that the primary factor in reconfiguration is solvent-based friction. The authors build upon their prior work in Grotz et al. to use atomistic molecular simulations to infer chain reconfiguration times for single-stranded nucleic acids from nanosecond FCS data.

The research is structured logically, and the results are presented clearly in this well-written

manuscript. The work is likely to be of broad interest as it provides new insights into the nanosecond-scale nucleic acid conformations and intramolecular interactions. Major and minor comments / questions are detailed below:

MAJOR:

Reweighting procedure: Why was variance of the smFRET distributions used to reweight the thermodynamic distribution of HCG structures? smFRET distribution widths should reflect burst duration (photon #) rather than structural reconfigurations, because all dynamics faster than the burst time are averaged out. Would it not be more appropriate to reweight only the mean of the computed FRET efficiency distributions, and not the variance, since variance cannot be inferred from the experiments performed here? If variance is not reweighted, what is the effect on the results? At the very least, the authors should justify and discuss this in a bit more detail.

MINOR:

1. Annotation text in many panels is too small; other sub-features of panels (Fig. 1a diagram of construct, for example) are too small.
2. The presentation of PMBioEn in Figure 2 is confusing. Either Figure 2 or the accompanying supplementary figure S5 should include the original HCG prediction for ssDNA. Currently, the symbol used to represent “PMBioEN” is not sufficiently distinguishable from the other markers and colors to be identified without reference to the caption.
3. Fig. 2b: Caption does not state the meaning of the vertical line (presumably RMS dye-dye distance). What are the significances (if any) of the colors?
4. Might be worth adding insets to 2d (or an SI figure) to illustrate that 0 is within fit error for all viscosity measurements.
5. A table of ground-truth mean values (and variances) which are used in the BioEn fitting process would be useful, in addition to the distributions shown in S5.

Reviewer #2 (Remarks on code availability):

The code at the provided URL, FRETICA, was introduced by the Schuler lab more than 10 years ago and is well-documented on GitHub; therefore, did not specifically review it here. HCG code is documented in previous publications.

Reviewer #3 (Remarks to the Author):

The current study applies FRET method to probe the chain dynamics of ssDNAs and ssRNAs. FRET and other experimental methods applied in the study have been well established for nucleic acids. Previous studies have shown that, at high ionic strength, ssDNA or ssRNA can fold/unfold rapidly because of the screening of electrostatic interactions. The current study just confirms this

observation, with not a lot of new discovery. It appears to me that publication in any form on Nat. Commun. would be premature at this time.

Response to the referees' comments

We thank the referees for their thorough reading of the manuscript and their very helpful and constructive comments, which have allowed us to further improve the manuscript as well as its presentation for a wide audience. We include our point-by-point response below (in blue) and have marked corresponding changes in the manuscript in red.

Referee #1:

The manuscript titled "Nanosecond chain dynamics of single-stranded nucleic acids" by Mark F. Nüesch, Lisa Pietrek, Erik D. Holmstrom, and colleagues presents a detailed study on the dynamics of single-stranded DNA (ssDNA) and RNA (ssRNA) oligonucleotides. This work combines single-molecule Förster resonance energy transfer (FRET), nanosecond fluorescence correlation spectroscopy (nsFCS), and nanophotonic (zero mode waveguide) enhancement to elucidate the rapid conformational dynamics and chain reconfiguration times of short ssNAs.

The authors aim to overcome the limitations in resolving the elementary chain dynamics of ssNAs, which are crucial for understanding their folding and function. By employing a sophisticated experimental setup, they manage to quantify the exceedingly rapid chain dynamics in the 10-ns range. Their findings suggest that these dynamics are predominantly influenced by solvent friction, with negligible internal friction, thereby providing a significant insight into the biophysical behavior of ssNAs.

The methodology employed in this study is highly advanced and innovative, combining cutting-edge techniques to probe the dynamics of ssNAs with unprecedented detail. The use of hierarchical chain growth and Bayesian inference for the analysis further strengthens the validity of their conclusions.

The results indicating the lack of detectable internal friction in ssNAs, regardless of the sequence composition, provide a new understanding of ssNA behavior in solution. This finding contrasts with the expected behavior based on the known structure and properties of ssNAs, offering a novel perspective on their dynamic properties.

In summary, the manuscript significantly advances our understanding of ssNA dynamics. It challenges existing notions about internal friction in biomolecules and provides a foundation for further research into the biophysical properties of ssNAs. The manuscript is written exceedingly clearly and present all the experimental and theoretical details required for justifying the made conclusions. I strongly recommend publication as is.

We thank the reviewer for the thorough analysis of our manuscript and the positive evaluation and detailed understanding of our work, particularly for appreciating the sophisticated methodology employed and the novel insights provided into the dynamics of single-stranded DNA and RNA.

Referee #2:

Nüesch et al. present experimental and computational results that indicate short single-stranded nucleic acid oligomers exhibit rapid chain dynamics with little internal friction, suggesting that the primary factor in reconfiguration is solvent-based friction. The authors build upon their prior work in Grotz et al. to use atomistic molecular simulations to infer chain reconfiguration times for single-stranded nucleic acids from nanosecond FCS data.

The research is structured logically, and the results are presented clearly in this well-written manuscript. The work is likely to be of broad interest as it provides new insights into the nanosecond-scale nucleic acid conformations and intramolecular interactions. Major and minor comments / questions are detailed below:

Thank you for your constructive feedback on our manuscript. We appreciate the positive remarks regarding the new insights and its structure and clarity.

MAJOR:

Reweighting procedure: Why was variance of the smFRET distributions used to reweight the thermodynamic distribution of HCG structures? smFRET distribution widths should reflect burst duration (photon #) rather than structural reconfigurations, because all dynamics faster than the burst time are averaged out. Would it not be more appropriate to reweight only the mean of the computed FRET efficiency distributions, and not the variance, since variance cannot be inferred from the experiments performed here? If variance is not reweighted, what is the effect on the results? At the very least, the authors should justify and discuss this in a bit more detail.

We thank the reviewer for the comment, which made us realize that this point was not sufficiently clear in the manuscript.

The widths of the FRET efficiency histograms indeed primarily reflect photon statistics since the rapid conformational dynamics average out during the burst, as we note in the manuscript (line 110-113):

“The widths of the transfer efficiency peaks of the unstructured ssNAs are close to the photon shot noise limit (Figure 1a, Supplementary Figure 1), indicating that inter-dye distance fluctuations are averaged out during the diffusion time of the molecules through the confocal volume of ~1 ms.”

The variance used in our reweighting process is not the one obtained from the widths of the FRET efficiency histograms but from a combined analysis of single-molecule FRET efficiencies and donor acceptor fluorescence lifetimes (introduced by Seidel & coworkers, ref. 37, and Gopich & Szabo, ref. 38), which allows us to assess the variance of the underlying transfer efficiency distribution, independent of shot noise. We now stress this aspect more explicitly in the revised manuscript (line 181ff):

“It is worth emphasizing that we use for this approach not the variance of the transfer efficiency histogram (Figure 1a), which is dominated by shot noise⁴⁶; rather, we use the variance of the transfer efficiency distribution that corresponds to the underlying distance distribution, which can be obtained from the deviations of the mean fluorescence lifetimes from the static FRET line (Figures 1e, 2a) if the distance dynamics are slower than the fluorescence lifetimes^{37, 38, 44}. Correspondingly, the reweighting takes into account not only experimental information on the average intramolecular distance but also on the variance of the distance distribution.”

Excluding the variance from reweighting significantly alters the resulting distributions and may result in an underestimate or overestimate of the width of the distance distribution of the conformations in the ensemble that is not compatible with the data shown in the lifetime vs. transfer efficiency plots (Figs. 1a and 2e). This point is illustrated by the distance distributions shown in Supplementary Figure 5 (for completeness, we also added the corresponding values of τ_r^{HCG} in Supplementary Table 4). For instance, $P(r)$ for rU₁₉ from HCG before reweighting is quite broad (gray distribution, left plot below), which leads to the clear discrepancy with the experimental data (gray point, right plot below). After reweighting, the distribution is narrower and in better agreement with the data (cyan distribution on

the left and cyan data points on the right). Note that the transfer efficiency is in good agreement with the experimental data in both cases. Without taking the variance into account for the reweighting, we do not obtain a distribution whose width is compatible with experiment.

Taking into account the variance from lifetime data is in fact an important advance in our approach, so we are grateful to the reviewer for raising this point, and we hope that our modifications clarify the issue.

MINOR:

1. Annotation text in many panels is too small; other sub-features of panels (Fig. 1a diagram of construct, for example) are too small.

Thank you for alerting us of these issues. The figures have been adjusted accordingly.

2. The presentation of PMBioEn in Figure 2 is confusing. Either Figure 2 or the accompanying supplementary figure S5 should include the original HCG prediction for ssDNA. Currently, the symbol used to represent “PMBioEN” is not sufficiently distinguishable from the other markers and colors to be identified without reference to the caption.

Thank you for pointing this out. Since the HCG predictions for ssDNA are not used for the analysis of the chain reconfiguration times (Figure 2), we had not included them. However, we agree that they may be of interest, so we now added the HCG distance distribution for ssDNA in Figure S5. Thank you also for noticing that the markers used for PMBioEn and HCGBioEn in Figure 2 were hard to distinguish; this has now been changed.

3. Fig. 2b: Caption does not state the meaning of the vertical line (presumably RMS dye-dye distance). What are the significances (if any) of the colors?

Thank you for alerting us of this omission. We now state that the vertical line corresponds to the RMS dye-dye distance.

The color code of the distance distributions is used in panel d. To clarify this point, we added “color code according to underlying distance distributions as shown in b” in the caption.

4. Might be worth adding insets to 2d (or an SI figure) to illustrate that 0 is within fit error for all viscosity measurements.

Thank you for noticing. We adjusted the shading of the confidence intervals to better visualize the uncertainty near zero.

5. A table of ground-truth mean values (and variances) which are used in the BioEn fitting process would be useful, in addition to the distributions shown in S5.

Thank you for pointing out the lack of such a table, which we have now incorporated (Supplementary Table 6).

Reviewer #2 (Remarks on code availability):

The code at the provided URL, FRETICA, was introduced by the Schuler lab more than 10 years ago and is well-documented on GitHub; therefore, did not specifically review it here. HCG code is documented in previous publications.

Referee #3:

The current study applies FRET method to probe the chain dynamics of ssDNAs and ssRNAs. FRET and other experimental methods applied in the study have been well established for nucleic acids. Previous studies have shown that, at high ionic strength, ssDNA or ssRNA can fold/unfold rapidly because of the screening of electrostatic interactions. The current study just confirms this observation, with not a lot of new discovery. It appears to me that publication in any form on Nat. Commun. would be premature at this time.

We agree that FRET and other methods have previously been used to study the behavior of nucleic acids, and that folding of ssDNA and ssRNA has been investigated. However, our study is not concerned with the folding of nucleic acids but with the exceedingly rapid chain dynamics of single-stranded nucleic acids in the nanosecond range, a process that, to our knowledge, has not previously been resolved experimentally.

Our work leverages advanced methodologies combining single-molecule Förster resonance energy transfer, nanosecond fluorescence correlation spectroscopy, and nanophotonic enhancement to achieve these measurements. Moreover, we present a comprehensive analysis combining hierarchical chain growth models, polymer models, and Bayesian inference to provide a detailed interpretation of our experimental data that goes beyond established techniques. This unique combination has allowed us to probe the conformational ensembles and dynamic behavior of ssNAs on previously inaccessible timescales and enabled us to reveal that internal friction in chain reconfiguration of these molecules is negligible. Although we do not probe nucleic acid folding itself, these insights are crucial for a deeper understanding of the molecular dynamics that influence RNA folding and function, aspects that are critical for many biological processes.

We believe that these contributions offer substantial new insights into the biophysics of nucleic acids, thus providing a strong justification for publication. We were pleased to see that the other reviewers agree with our perception, and we hope that by better explaining the focus of our work, we can now also convince reviewer #3 of its relevance and timeliness.

REVIEWERS' COMMENTS

Reviewer #2 (Remarks to the Author):

Many thanks to the authors for their clarity and attention to detail in their response and revisions. The addition of an explanation for the distance-distribution-based variance reweighting resolves the previous major question, and the changes to figures and text to address minor comments are more than satisfactory. This work represents an important advance in quantitatively connecting simulated and experimental measurements of single-stranded nucleic acid reconfiguration dynamics, and therefore is expected to be of broad interest to the readership of Nature Communications. Happy to recommend this for publication as-is, and looking forward to seeing it in print.